# *Petiveria alliacea* Reduces Tumor Burden and Metastasis and Regulates the Peripheral Immune Response in a Murine Myeloid Leukemia Model

**DOI:** 10.3390/ijms241612972

**Published:** 2023-08-19

**Authors:** Natalia Murillo, Paola Lasso, Claudia Urueña, Daniel Pardo-Rodriguez, Ricardo Ballesteros-Ramírez, Giselle Betancourt, Laura Rojas, Mónica P. Cala, Susana Fiorentino

**Affiliations:** 1Grupo de Inmunobiología y Biología Celular, Pontificia Universidad Javeriana, Bogotá 110211, Colombia; natalia.murillo@javeriana.edu.co (N.M.); plasso@javeriana.edu.co (P.L.); curuena@javeriana.edu.co (C.U.); r.ballesteros@javeriana.edu.co (R.B.-R.); gybetancourt@javeriana.edu.co (G.B.); rojasl.a@javeriana.edu.co (L.R.); 2Metabolomics Core Facility—MetCore, Vicepresidency for Research, Universidad de Los Andes, Bogotá 111711, Colombia; d.pardorodriguez@uniandes.edu.co (D.P.-R.); mp.cala10@uniandes.edu.co (M.P.C.)

**Keywords:** acute myeloid leukemia, murine model, hematologic parameters, *Petiveria alliacea*, metastasis

## Abstract

The poor response, adverse effects and drug resistance to treatment of acute myeloid leukemia (AML) have led to searching for safer and more effective therapeutic alternatives. We previously demonstrated that the alcoholic extract of *Petiveria alliacea* (Esperanza) has a significant in vitro antitumor effect on other tumor cells and also the ability to regulate energy metabolism. We evaluated the effect of the Esperanza extract in vitro and in vivo in a murine model of AML with DA-3/ER-GM cells. First, a chemical characterization of the extract was conducted through liquid and gas chromatography coupled with mass spectrometry. In vitro findings showed that the extract modulates tumor metabolism by decreasing glucose uptake and increasing reactive oxygen species, which leads to a reduction in cell proliferation. Then, to evaluate the effect of the extract in vivo, we standardized the mouse model by injecting DA-3/ER-GM cells intravenously. The animals treated with the extract showed a lower percentage of circulating blasts, higher values of hemoglobin, hematocrit, and platelets, less infiltration of blasts in the spleen, and greater production of cytokines compared to the control group. These results suggest that the antitumor activity of this extract on DA-3/ER-GM cells can be attributed to the decrease in glycolytic metabolism, its activity as a mitocan, and the possible immunomodulatory effect by reducing tumor proliferation and metastasis.

## 1. Introduction

Acute myeloid leukemia (AML) is a type of cancer that originates from the neoplastic transformation of hematopoietic stem cells, due in part to a series of mutations that lead to a block in differentiation and, therefore, to an imbalance between cell proliferation and maturation, resulting in inefficient hematopoiesis [1,2]. To date, the first-line treatment is based on chemotherapy; however, the complete remission rate is still not optimistic, especially in elderly patients, and the recurrence rate after complete remission is still high [3]. The resistance of leukemic cells to chemotherapy becomes the main obstacle in the treatment of AML, which underlines the need to find more efficient and less toxic therapeutic alternatives [4].

Several factors influence the success of antileukemic therapies. Publicly recognized mechanisms of resistance to common treatments in AML are: (1) proteins and enzymes related to drug resistance such as P-gp; (2) genetic alterations in FLT3/WTN1 or the RAS family; (3) miRNA alterations in drug resistance, and (4) aberrant activation of signaling pathways related to metabolic adaptation leading to survival and proliferation, for instance, PI3K/AKT/mTOR [5]. AMLs are particularly sensitive to glycolytic metabolic modulators [6] or mitocans that target tumor mitochondria. In addition, the use of regulators of glycolytic metabolism in both cell lines and primary blasts obtained from patients, increases sensitivity at much lower doses than those affecting normal cells [7]. Mitocans can also promote immune response activation through inflammasome activation and induction of immunogenic death and even enhance immunotherapy [8].

Our group has been working on the evaluation of the antitumor effect of different plant extracts in vitro and in vivo, in breast cancer and murine melanoma models [9,10]. *P. alliacea* has been used by traditional medicine for the treatment of cancer [11,12], and although the cytotoxic activity of the ethanolic extract against leukemia cells is not very high [13], it was possible to evidence its antitumor activity against various leukemic cell lines such as K562, HL60MX, NB4, SUP-B15, Jurkat, U937, and RS4 when the secondary metabolites of the plant are concentrated in organic extracts of ethyl acetate [14,15,16]. We have shown that the Anamu extract regulates energetic metabolism by altering the expression of enzymes involved in glycolysis decreasing glucose uptake, and increasing lactate secretion [17]. Then, it also alters ßF1ATPase expression, decreasing mitochondrial respiration and intracellular ATP levels [9,18]. Additionally, the extract, in combination with chemotherapeutics or in monotherapy, shows activity in cell lines or the primary leukemic blast [15].

Currently, part of new drug discovery begins with in vitro or in silico research, but the predictive value of this data is limited by the complexity of whole-body systems, therefore, the use of animal models is necessary for the evaluation of new therapeutic strategies. The fundamental advantage of animal models is that relationships between the microenvironment and mechanisms of the immune system in cancer can be assessed under controlled experimental conditions, which can then be used to predict clinical outcomes in humans [19]. The mouse (*Mus musculus*) is the most widely used animal model of human disease due to its genetic and physiological similarity to humans [20,21,22]. Many factors have limited the development and use of animal models of leukemia. The liquid nature of hematologic malignancies, the complex microenvironment from which tumor cells arise, the large number of genetic alterations [23], and differences between murine and human hematopoietic homeostasis, as the lymphocyte predominance in peripheral blood of mice vs. neutrophil predominance in humans, are some of these limitations [24]. Murine models were implemented to study AML, beginning with carcinogen-induced transplantable models, and later transgenic, xenograft, and mosaic models [19], and in spite of being far from the disease physiopathology, they have contributed to the development of modern anti-leukemic chemotherapy, such as cytarabine. An adaptive transfer model of in vitro modified murine hematopoietic stem or progenitor cells (HSPC), using genetic edition followed by tail intravenous (i.v.) transplantation, allows mimicking the leukemic phenotype in a healthy microenvironment and exploring the role of the microenvironment and immune response related to the response to treatment [25].

Considering that the availability of a murine model of AML would provide a useful tool for the evaluation of new therapeutic strategies, the objective of this work was to standardize in our hands an orthotopic model of murine AML and to evaluate the in vivo activity of a standardized *Petiveria alliacea* extract, previously evaluated in other tumor models [14,17,26]. We used the myeloid DA-3/ER-GM cell line isolated from a BALB/c mouse infected with the Moloney murine leukemia virus (MoMLV) previously published [27,28]. The DA-3/ER-GM cell line shows a rapid dissemination to the spleen and bone marrow, inducing anemia, thrombocytopenia, and the presence of blasts in peripheral blood, which are easily evaluated for monitoring the disease progression and the response to treatment [27].

## 2. Results

### 2.1. Chemical Characterization of Esperanza Extract

HPLC-ESI-QTOF-MS and GC-EI-QTOF-MS in both positive and negative ionization modes were used to characterize the chemical compounds of the Esperanza extract. The chromatograms and major compounds identified for the analytical platforms are shown in Appendix A. All the annotated metabolites are included in Appendix A, along with the relevant chemical variables and the identification level.

A total of 155 compounds were annotated in the extract, distributed as follows across several compound classes: amino acids, peptides, and derivatives made up 29.29%, followed by fatty acids (18.47%), organooxygen compounds (10.82%), benzenes and their derivatives (8.91%), imidazopyrimidines (4.45%), prenol lipids (3.82%), and cinnamic acids and derivatives (3.18%) (Appendix A). Other recognized groups with a lower participation in the composition were indoles, organooxygen compounds, and phenols, each accounting for 2.24%. Additionally, flavonoids, pirroles, hydroxy acids, diazines, and pyridines each contributed 1.27% to the total composition (Appendix A).

Regarding the contribution established by relative abundance, certain compounds analyzed by GC/MS were citric acid (12.97%), glutamic acid (10.82%), aspartic acid (7.09%), oxalic acid (6.47%), succinic acid (6.00%), palmitic acid (6.14%), glucaric acid (4.50%), linolenic acid (1.71%), and benzoic acid (1.69%) (Appendix A). The characterization performed by LC/MS found compounds such as amino-methylenehexanoic acid (6.44%), proline betaine (6.44%), phenylacetaldehyde (2.81%), methylbenzaldehyde (2.58%), germacradiene-acetoxy-diol (2.04%), indoleacrylic acid (1.93%), corchorifatty acid f (1.64%), oxo-octadecatetraenoic acid (1.42%), fulgidic acid (1.11%), and auxin b (1.01%), representing the major contributions (Appendix A).

### 2.2. In Vitro Activity of Esperanza Extract against DA-3/ER-GM Cells

Previously, the alcoholic Anamu extract derived from the *Petiveria Alliacea* was shown to have a significant in vitro antitumor effect in breast adenocarcinoma and other leukemia cells [9,14,15]. Therefore, we evaluated the in vitro effect of the standardized aqueous extract “Esperanza” against DA-3/ER-GM cells and other different cell lines. First, we evaluated the cell morphology of DA-3/ER-GM cells through cytospin preparations. DA-3/ER-GM cells have a slightly basophilic cytoplasm, poor chromatin, nucleoli, 4:1 nucleus:cytoplasm ratio, and some cytoplasmic granules, which is consistent with the description of an immature cell with a blastic appearance (Figure 1A). The Esperanza extract does not present a marked cytotoxic activity in any of the cell lines evaluated (DA-3/ER-GM, U937, MCF-7, MDA-MB-468 and 3T3), except on 4T1, where the IC_50_ is less than 50 µg/mL (Table 1).

However, when analyzing the effect of the Esperanza extract on the proliferative capacity of DA3/ER-GM cells, a dose- and time-dependent reduction in proliferation was observed (Figure 1B). Likewise, glucose uptake measured with the 2-NBDG (2-(N-(7-Nitrobenz-2-oxa-1,3-diazol-4-yl)Amino)-2-Deoxyglucose) probe decreased mainly at 12 h with Esperanza extract and between 12 and 24 h with the LY294002 inhibitor, as expected (Figure 1C and Appendix A). Intracellular reactive oxygen species (ROS) concentrations measured with the H_2_DCFDA probe increased significantly in a dose- and time-dependent manner with Esperanza versus Doxorubicin (Figure 1D and Appendix A). Treatment with the extract significantly reduced glucose uptake and increased intracellular ROS. This supports previous results [9,18], and suggests that Esperanza extract may modulate tumor metabolism inducing a cytostatic effect.

### 2.3. Standardization of the AML Model

To define the optimal DA-3/ER-GM cell number for murine AML development and to study the effect of future treatments, different numbers of DA-3/ER-GM cells were transplanted as described (Figure 2A). On the third day, peripheral blood smears of all mice led to evidence of blast presence, which increased to 20% in mice inoculated with 1 × 10^6^ and 5 × 10^5^ tumor cells. These animals died on the 11th day. In the group of mice inoculated with 5 × 10^4^ cells, peripheral blasts gradually increased until day 15 post-inoculation. Transplantation of 5 × 10^3^ and 5 × 10^2^ cells generated late disease development (Figure 2B).

Analysis of hematological parameters showed that total red blood cell count (RBC; Figure 2C), hemoglobin (HGB; Figure 2D), hematocrit (HCT; Figure 2E) and platelet count (PLT; Figure 2F) were decreased in all mice groups where the parameter could be analyzed compared to basal levels (Day 0), possibly as a consequence of the progressive and abnormal accumulation of blasts in bone marrow that leads to inefficient hematopoiesis [29]. In contrast, an increase in the white blood cell (WBC; Figure 2G) count was found at the end of the analysis (Day 12), which could be related to the increase in peripheral blood blasts. As endpoint criteria, the animals presented hunched posture, lack of grooming and marked piloerection (Appendix A), lethargy, 10% weight loss, and pale mucous membranes, possibly because of anemia. When mice presented these endpoint criteria, the animals were euthanized and necropsied. Pallor of the liver and mucous membranes, hepatosplenomegaly, and hemorrhagic areas in the lungs were observed (Appendix A).

### 2.4. Histopathological Analysis of Animals with AML

As previously described [27], histopathological analyses showed extensive tumor infiltration in all organs (Figure 3). In the spleen, a dense and diffuse infiltrating lymphoproliferative process of the splenic stroma was evidenced, causing the loss of the organ structure (Figure 3). In addition, few foci of mature lymphocytes (black arrows), cells with two or three times the size of the erythrocyte (blasts, white arrows), poorly defined cytoplasmic boundaries, moderate anisokaryosis, and cellular pleomorphism with polyhedral oval shapes, lumpy chromatin, vacuolated, and evident nucleoli were found. On the other hand, moderate lymphocytolysis and extramedullary hematopoiesis were evidenced by the presentation of megakaryocytes (arrowhead) and a few immature cells of the erythroid line (erythroblasts) (Figure 3).

In the bone marrow, a diffuse and marked proliferative process was observed, resulting in the loss of cell cords of the myeloid and erythroid lineages, anisocytosis, and anisokaryosis with cellular pleomorphism, indicative of marrow failure (arrows) (Figure 3). Loss of linearity of the hepatic cords was found in the liver, with marked retention of inflammatory and neoplastic cells (blasts) in hepatic sinusoids (arrows), accompanied by neoplastic cells in portal triads and slight foci of extramedullary hematopoiesis, the presence of megakaryocytes (arrowhead), and slight glycogenic changes inside the hepatocytes (Figure 3). Finally, generalized congestive changes were found in the lungs, acute alveolar hemorrhage foci (arrows) and slight retention of inflammatory and neoplastic cells in the alveolar septa and vascular lumen (arrowhead) (Figure 3). All the above is correlated with the metastasis sites of the tumor cells, and what was observed macroscopically at the time of necropsy.

### 2.5. In Vivo AML Model and Treatment Evaluation with the Esperanza Extract

Subsequently, we evaluated the in vivo effect of Esperanza in the murine AML model. The mice were inoculated with 5 × 10^4^ DA-3/ER-GM cells and then treated five times per week with Esperanza extract, starting 3 days after the inoculation, after verification of the presence of blast cells in the periphery, and until the end of the experiment (Figure 4A). From day 7 to day 17 post-inoculation, treated mice showed a 40% decrease in the number of circulating blast cells compared to the control group (Figure 4B). Likewise, on day 17 post-inoculation, it was found that the Esperanza extract maintained the levels of RBC (Figure 4C), HGB (Figure 4D), HCT (Figure 4E) and PLT (Figure 4F) within the normal range compared to control animals. In contrast, no differences were found in the WBC count (Figure 4G) at the end of the model (Day 17). On the other hand, a significant decrease in the number of spleen-infiltrating blasts was found, as was a decreasing trend in the bone marrow and lung of the treated mice compared to the control group (Figure 4H). Furthermore, at the end of the experiment, Esperanza-treated mice exhibited increased plasma concentrations of IL-10, IL-17, IFN-γ, IL-6, IL-4, and IL-2 compared to controls (Figure 4I).

Histopathological analysis in the liver of Esperanza-treated mice showed a mild infiltrate of immature lymphocytes around the portal triads and the central vein (arrows), and in the hepatic sinusoids (arrowhead), in contrast to the control mice, where a greater infiltrate of immature lymphocytes was found around the portal triads (arrows). On the other hand, lung analyses showed moderate generalized congestive changes (arrows), areas of alveolar atelectasis (arrowhead), and slightly hyperplastic changes in BALT in control mice, but not in animals treated with Esperanza. Bone marrow was normal for both groups with a heterogeneous population of lymphocytes and megakaryocytes (arrows), in spite of the presence of tumor cells (Figure 5).

## 3. Discussion

The low cure and survival rates of AML patients highlight the importance of developing animal models. In this work, we have reproduced a model of non-lymphoid neoplasia in BALB/c mice by transplantation of GM-CSF-induced, MoMLV-autoactivated DA-3/ER-GM cells, as previously reported [27]. This model exhibits a rapid growth of circulating blasts that are also localized in the bone marrow and spleen. In fact, it was documented that one of the main host-associated factors that maintain the leukemic condition in retrovirus-infected animals had high levels of cytokines and growth factors, such as IL-3 and GM-CSF [30]. Since this line expresses GM-CSF endogenously, this could explain its rapid growth, as seen in this work. The pattern of tumor cell growth and organ dissemination observed after i.v. inoculation of DA-3/ER-GM cells resembles that of AML. In fact, circulating blasts were observed in peripheral blood on the third day post-inoculation, which settled in the bone marrow, generating a reduction of red blood cells, hemoglobin, hematocrit, and platelets.

We took advantage of this model to evaluate the in vivo activity of a standardized aqueous extract of *Petiveria alliacea*, which we named “Esperanza”. Anamu leaves and root powder were traditionally as infusions to treat rheumatism, spasmodic emesis, fever, intestinal parasites, and tumors, including leukemias and breast cancer [31]. An ethyl acetate-derived Anamu extract was previously shown to disrupt the cell cytoskeleton, inducing cell cycle arrest in the A375 cell line [17], as was demonstrated in the neuroblastoma cell line, SH-SY5Y, treated with disassembly of microtubules and inhibition of neurite outgrowth, neuroblastoma cell proliferation, and MAP kinase tyro-sine dephosphorylation by dibenzyl trisulphide [32]. This activity is accompanied by dysregulation of proteins related to energy metabolism, the cytoskeleton, and cell adhesion [17]. These findings were subsequently confirmed in murine 4T1 breast cancer cells, in which decreased glycolysis, mitochondrial respiration, lactate, and ATP production were also observed, along with reduced expression of β-F1ATPase [18]. There is no doubt that other compounds, very poorly represented in the extract, may explain the traditional use of this plant for several decades. Some of these minor metabolites, not previously described in Anamu, have important antitumor activity. Oxalic acid (glycolysis regulator), Malonic acid (mitochondrial uncoupler, inhibits ATP synthase), Benzoic Acid (inhibits histone deacetylases (HDAC); isoleucine (inhibits PTEN ubiquitination and induce apoptosis by disrupting the mitochondrial membrane); Glyceric acid (inhibits enolase that catalyses the conversion of 2-phosphoglycerate (2-PG) to phosphoenolpyruvate (PEP), the ninth and penultimate step of glycolysis); Glutamic acid (interfere with the metabolic process of l-glutamine acting as anticancer agent) and Corchorifatty acid F (it is found in *Sternbergia clusiana* and has an antiproliferative effect) [33,34,35,36,37,38,39,40]. One of the major compounds found in Esperanza was citric acid. This is one of the most effective inhibitors of glycolysis, which inhibits the phosphofructokinase enzyme blocking glycolysis at the start; inhibits the pyruvate dehydrogenase enzyme complex; and also inhibits the succinate dehydrogenase enzyme of Krebs cycle [41].

These and other activities of Anamu were recently reviewed [14], confirming the enormous potential of this plant in cancer treatment due to the intrinsic synergy of its components, which allow it to act as both a mitochondrial uncoupler and a glycolytic inhibitor. These molecular targets have been implicated in the effective killing of chemotherapy-resistant primary AML cells, which appear to be more sensitive to mitochondrial damage than other tumor blasts derived from solid tumors [7]. The impairment of energy metabolism in terms of oxidative phosphorylation and tumor cell glycolysis caused by the extract could lead to tumor cell death by energy deprivation.

In our hands, animals treated with Esperanza showed a clear reduction in spleen metastases and a decrease in peripheral blood blasts. In addition, the hematological parameters (RBC, HGB, HTO and PLT) were found within normal ranges on day 17 post-tumor transplantation compared to animals treated with PBS (Figure 4). Changes in cellular metabolism are described as characteristic of cancer, and metabolic targeting is considered a possible therapeutic strategy in AML [42]. Cancer cells often rely more on glycolysis and have higher energy requirements compared to normal cells that mainly employ oxidative phosphorylation (OXPHOS) for energy in the presence of oxygen [43]. Glycolysis fulfills the bioenergetic needs of the tumor and is crucial for proliferation, survival, aggressiveness, and resistance to conventional therapies in AML. Several glycolytic inhibitors have been shown to decrease proliferation, induce apoptosis, and reverse the resistant phenotype against different anthracyclines used in AML. In murine models of AML, glycolytic inhibitors such as 3-BrOP, 2DG, and Sorafenib have caused a reduction in the number of circulating blasts, a reduction in metastasis, and an increase in survival in treated animals during disease progression, supporting that glycolytic metabolism plays a key role in leukemogenesis [44]. It has been reported that Anamu not only modulates metabolism through decreased glucose uptake, but also induces apoptosis through a mitochondria-dependent pathway, upregulates HSP70 expression [26], affects the Beta-F1-ATPase pump causing ATP depletion [18], and increases reactive oxygen species in different tumors. These different mechanisms of action of the extract could affect the proliferation and metastasis of leukemic cells, as we observed in vitro and in vivo.

Regarding the immunomodulatory role of Esperanza, we observed an increase in IL-10, IL-4, IFN-γ, IL-17, IL-6 and IL-2 concentrations in the plasma of treated animals. Although each of these cytokines is subject to a different regulatory mechanism, the orchestration of this response could be directed by IL-10, which plays an important role in the control of inflammation and cancer. In fact, IL-10 deficiency both in mice and humans is related to inflammatory bowel disease and cancer [45,46]. It was observed that IL-10 application was efficacious in the effector phase of tumor rejection, whereas application during tumor vaccination failed to significantly enhance tumor rejection [47]. IL-10 enhances the proliferation of CD8+ T cells stimulated through the T cell receptor (TCR) using anti-CD3 monoclonal antibodies (mAbs) and functionally replaces IL-2 in the proliferation of CD8+ T cells stimulated with anti-CD3 [48]. In fact, tumor cells transformed to express IL-10 are rejected in immunocompetent hosts [49] and transplantable tumors injected into genetically transformed mice to express human IL-10 in myeloid cells were rejected, depending on the presence of CD8+ T cells in the host [50]. Furthermore, treatment of syngeneic mouse tumors with recombinant human IL-10 induced CD8+ T-cell-dependent tumor rejection [51].

Emmerich et al. found that continuous IL-10 treatment leads to CD8+ T cell dependent rejection of large endogenous breast cancers in Her2 transgenic mice by eliminating lung metastases. IL-10 treatment also led to the generation of a tumor-protective immune memory capable of killing transplanted tumors, even up to 8 months after initial tumor rejection. This long-lasting tumor immunity is likely mediated by an IL-10 signaling pathway specific for intratumoral CD8+ T cells, IL-10-induced STAT1 and STAT3 phosphorylation, and IFN-γ expression in intratumoral CD8+ T cells [52]. In support of this concept, the in vivo maintenance of antiviral memory CD8+ T cells also requires STAT3 and IL-10 [53] and, additionally, intratumoral IL-10 injections induce an increase of IFN-, IL-4 and IL-18, improving the cytotoxic activity of CD8+ T cells [54]. In fact, it is under discussion whether IL-10 increases antitumor immunity in humans [55].

We have previously observed that the aqueous fraction of Anamu is immunomodulatory, inducing IL-10, among other cytokines, and a partial activation of DC, which points to a reduction in the inflammatory response [56]. The mechanisms that regulate the production of cytokines in response to treatment with plant extracts are not yet well understood, but receptors such as the aryl hydrocarbon receptor (AHR) could be involved. In fact, recent studies identified the effects of AHR signaling on multiple aspects of the immune response, including IL-10 [57] and IL-17 production [58]. Although IL-10 can be produced by multiple cell populations, it is possible that NK cells are primarily involved [59]. Interestingly, it has been described that IL-4 can act as a selective inhibitor of AML cell growth and survival, but does not affect normal hematopoietic stem cells. IL-4 selectively induces apoptosis in these tumor cells in a Stat6-dependent manner in vitro and in vivo, therefore, treatments that induce upregulation of these cytokines, such as Esperanza extract, could be considered therapeutic alterations [60]. Additionally, we observed that the extract may have an immunomodulatory effect by increasing cytokines such as IL-4, which may act as a selective inhibitor of AML cell growth and survival [60,61].

What are the metabolites involved in this stimulation? We do not know, but since the extract is aqueous, the role of the primary metabolites present in the preparation cannot be ruled out [62]. Polysaccharides and some cell-derived peptides have been lesser studied, however, they could be participating not only in the reduction of the tumor load or in the cytostatic activity of this plant, but also in the activation of the antitumor immune response.

## 4. Materials and Methods

### 4.1. Tumor Cell Lines and Culture Conditions

The DA-3/ER-GM cells were kindly provided by Prof. Jose Arteaga (Departament of Basic Science, Medical School, Universidad Industrial de Santander, Bucaramanga, Colombia). DA-3/ER-GM cells were cultured in RPMI-1640 medium (Eurobio, Toulouse, France) with 10% heat-inactivated fetal bovine serum (FBS), 2 mM L-glutamine, 100 U/mL penicilin, 100 μg/mL streptomycin, 0.01 M HEPES buffer, 1 mM sodium pyruvate (Eurobio), and 50 pg/mL gentamycin (Gibco, Grand Island, NY, USA) and cultivated at 37 °C in 5% CO_2_. Evaluation of cell morphology was performed with 2 × 10^5^ cells in Cytospin preparations and stained, using the Wright stain, prior to photographing using 100× magnification.

### 4.2. Animals

Female BALB/cAnNCrl young (aged 6 to 10 weeks) mice were housed at the animal facilities from the Pontificia Universidad Javeriana (PUJ, Bogotá, Colombia) following the established protocols of the Ethics Committee of the Faculty of Sciences and national and International Legislation for Live Animal Experimentation (Colombia Republic, Resolution 08430, 1993; National Academy of Sciences, 2010). The protocol was approved by the animal experimentation committee of PUJ (FUA-104-21).

### 4.3. Plant Material and Preparation

Leaves and stems of the *Petiveria alliacea* were collected in the wild in the municipality of Lerida (Department of Tolima, Lerida, Colombia) at 255 m above sea level and the coordinates are E 01242297 N 01042009. The plant material was identified by Carlos Alberto Parra of the Colombian national herbarium with the voucher for specimen number COL193. After receiving the material, the leaves were separated from the rest of the material, and serial washings were carried out. The leaves were subjected to an extraction procedure by infusion with boiling water for a previous established time in a ratio emulating the traditional use. Subsequently, this extract was subjected to lyophilization. Three batches were manufactured and characterized following the protocol described above, ensuring chemical consistency between batches followed Food and Drug Administration (FDA) regulations (Appendix A).

Chromatographic profiling was performed by Ultra Performance Liquid Chromatography, coupled with Photodiode Array Detection (UPLC-PDA) at 254 nm wavelength, showing the presence of different peaks of high to medium polarity observed in Rt = 1.0 to 13.0 min (Appendix A). The final extract was certified according to USP (United States Pharmacopeia) and WHO (World Health Organization) guidelines; the production of Anamu extract (Esperanza) was performed under GMP conditions in Labfarve Laboratories with the subsequent physicochemical and microbiological certification before starting the experiment.

The P2Et standardized extract, used as anti-oxidant and anti-tumor control [10,63], was produced under good manufacturing practice (GMP) conditions and chemically characterized as previously described [64,65,66].

### 4.4. Chemical Characterization of the Experanza Extract

#### 4.4.1. Chemical Composition through RP-LC-QTOF-MS

Five milligrams of the lyophilized extract were suspended in 200 µL of a MeOH/H_2_O mixture (25:75), vortexed for 15 min, subjected to ultrasonication for 10 min, and vortexed again for 5 min. The resulting extract was centrifuged at 16000 rpm for 10 min at 25 °C, and the supernatant was filtered through 0.22 µm filters for subsequent analysis by RP-LC-QTOF-MS.

The analysis was performed using an Agilent Technologies (Santa Clara, CA, USA) 1260 Liquid Chromatography system, coupled to a quadrupole time-of-flight mass analyzer Q-TOF 6545 with electrospray ionization. Five microliters of the sample were injected into a C18 column (InfinityLab Poroshell 120 EC-C18, 100 × 3.0 mm, 2.7 µm) at 30 °C and eluted using a gradient composed of 0.1% (*v*/*v*) formic acid in type I water (Phase A) and 0.1% (*v*/*v*) formic acid in acetonitrile (Phase B) with a constant flow rate of 0.4 mL/min. The gradient started at 2% Phase B and gradually increased to 30% Phase B over 15 min. Subsequently, the gradient was increased to 98% Phase B in two minutes and held for another two minutes. Finally, the gradient was decreased after one minute to 20% Phase B and held for an additional five minutes until equipment reconditioning. Mass spectrometry detection was initially performed in positive and negative ESI mode with full scan from 50 to 1100 *m*/*z*. Additionally, three mass spectra were taken iteratively for 20 and 40 eV energies.

#### 4.4.2. Chemical Composition through GC-EI-QTOF-MS

Ten milligrams of the lyophilized extract were suspended in 20 µL of methoxyamine hydrochloride in pyridine (15 mg/mL) and vortex-mixed for 5 min. The extract was incubated for 16 h at room temperature in the dark. The silylation process was initiated by adding 20 µL of bis(trimethylsilyl)trifluoroacetamide with 1% trimethylchlorosilane and then incubated at 70 °C for 1 h. The samples were diluted with 50 µL of internal standard (methyl heptadecanoate-d33, 5 µg/mL). The analysis was performed on an Agilent Technologies 7890B GC system coupled to a 7250 QTOF mass spectrometer system (Agilent Technologies). The derivatized extract was injected (1 µg/mL) with a split ratio of 30:1 onto an HP-5MS capillary column (30 m × 0.25 mm; 0.25 µm) (Agilent Technologies) at a constant gas flow (helium) of 0.7 mL/min. The injector temperature was 250 °C. The temperature gradient was kept at 60 °C for 1 min and then programmed to 320 °C at 10 °C/min. Mass spectra were recorded at 70 eV in full scan mode with values ranging from 50 to 600 *m*/*z*. The transfer line, filament source, and quadrupole temperature were fixed at 280, 230, and 150 °C, respectively.

#### 4.4.3. Data Processing

The LC-MS raw data were processed using the Agilent MassHunter Profinder software version 10.0 (B.10.0, Agilent Technologies) to extract molecular features for deconvolution, alignment, and integration. The GC-MS data was treated with deconvolution and identification of metabolites using the Agilent MassHunter Unknowns Analysis software version 10.0 and the Fiehn and NIST libraries. Retention time alignment was then performed in the Agilent Mass Profinder Professional software version 10.0, and the results were exported to the Agilent MassHunter Quantitative software version 10.0 for data integration.

#### 4.4.4. Metabolite Identification

For LC-MS, the chromatographic characteristics were annotated incorporating diverse criteria: exact mass matching with the values in databases using the CEU Mass Mediator annotator and the available *m*/*z* values in http://ceumass.eps.uspceu.es (acceded on 15 June 2023), generation of theoretical formulas utilizing isotopic distributions, verifying retention times, and checking for the formation of adducts. Moreover, iterative MS/MS data were cross-referenced with the spectra data from MS-DIAL 4.80, http://prime.psc.riken.jp/compms/msdial/main.html (acceded on 15 June 2023) and the Lipid Annotator software version 1.0. Manual interpretation of the MS/MS spectrum was also undertaken. The metabolites derived from GC-MS analysis were annotated using the Fiehn version 2013 and NIST libraries, along with the MassHunter Personal Compound Database and Library Manager Software version 8.0 (B.08.00, Agilent Technologies). The identification of metabolites by LC-MS and GC-MS was executed based on a 4-level confidence system for high-resolution mass spectrometry analysis, following the parameters previously described [66].

### 4.5. Reagents

LY294002 (Bio-Techne Corporation, Danvers MA, USA), a compound derived from the naturally occurring biflavonoid quercetin, which inhibits PI3K activity via competitive inhibition of an ATP binding site [67,68] was freshly prepared in dimethyl sulfoxide (DMSO) and used at the final concentrations indicated. Doxorubicin (MP Biomedicals, Santa Ana, CA, USA) was prepared in DMSO and used as pro-oxidant control.

### 4.6. In Vitro Cytotoxicity Assays

Esperanza cytotoxic effect on DA-3/ER-GM, U937, 4T1, MCF-7, MDA-MB-468 tumor cells, and mouse fibroblast cell line 3T3 was evaluated using methylthiazol tetrazolium (MTT) assay (Sigma-Aldrich, Saint Louis, MO, USA), as previously reported [17]. The half maximal inhibitory concentration (IC_50_) value was calculated using GraphPad Prism version 8.1.1 for Mac OS X statistics software (GraphPad Software, San Diego, CA, USA).

### 4.7. Proliferation Assay

1 × 10^5^ DA-3/ER-GM cells were seeded in 12-well plates with H_2_O (negative control), different concentrations of Esperanza extract (23, 46, 92 and 184 µg/mL) for 12, 24, 48, 72 and 96 h and counted by trypan blue. The exponential (Malthusian) growth formula Y = Y0 × exp(k × x) of the Software GraphPad Prism version 9 for Mac OS X was used to calculate the doubling time.

### 4.8. Glucose Uptake Assay

For evaluation of glucose uptake, 3 × 10^5^ cells were seeded on 12-well plates and incubated for 12 and 24 h with 23 and 46 µg/mL of the Esperanza extract, 20 µM of LY294002 (control of decreased glucose uptake), H_2_O (extract vehicle) or DMSO (LY294002 vehicle). After treatments, cells were removed, PBS washed, and resuspended in 40 µM of 2-NBDG (2-(N-(7-Nitrobenz-2-oxa-1,3-diazol-4-yl)Amino)-2-Deoxyglucose) (Invitrogen Molecular Probes) prepared in RPMI 1640 without phenol red. Then, cells were incubated for 30 min at 37 °C and washed with cold PBS 1X. Live versus dead cell discrimination was performed labeling with LIVE/DEAD Fixable Aqua (Life Technologies, Thermo Scientific, Eugene, OR, USA). Immediately, samples were acquired by Cytek Aurora Cytometer (Cytek Biosciences, Fremont, CA, USA) and analyzed with FlowJo v10.8.1 software (BD Life Sciences, Franklin Lakes, NJ, USA). Experiments were performed in triplicate and the results were expressed as mean ± SEM.

### 4.9. ROS Measurement

To evaluate ROS production, 1.5 × 10^5^ cells were plated in 12-well plates and treated with 23, 46, 92 and 184 µg/mL of the Esperanza extract, 0.02 µM of Doxorubicin (pro-oxidant control), P2Et extract (anti-oxidant control), H_2_O (Esperanza vehicle), ethanol (P2Et vehicle) or DMSO (Doxorubicin vehicle) for 12 and 24 h. Cells were stained with 1 µM H_2_DCFDA (2′,7′-dichlorodihydrofluorescein diacetate) (Sigma Aldrich, Saint Louis MO, USA) for 40 min at 37 °C, followed by Propidium Iodide (PI) (Sigma-Aldrich). Each sample was then acquired using a Cytek Aurora Cytometer (Cytek Biosciences, Fremont, CA, USA) and analyzed with FlowJo v10.8.1 software (BD Life Sciences). Experiments were performed in triplicate and the results were expressed as mean ± SEM.

### 4.10. Induction of AML in Mice

One day before the experiment, 20 µL of peripheral blood drawn from the tail vein were collected to evaluate hematological parameters. The next day, five groups of mice (*n* = 5) were inoculated intravenously (i.v.) by lateral tail-vein injection with 5 × 10^2^, 5 × 10^3^, 5 × 10^4^, 5 × 10^5^ or 1 × 10^6^ of DA-3/ER-GM cells. All mice were monitored three times a week for body weight and signs of illness (hunched posture, ruffled fur, lethargy, and pale mucous membranes). Additionally, hematological parameters were analyzed with a blood count using the Mindray BC-5150 (Mindray Bio-Medical Electronics Co., Ltd., Shenzhen, China) and a peripheral blood smear Wright’s stained using an automated stainer (Aerospray Stainer, Wescor, Logan, UT, USA). Mice were euthanized by CO_2_ inhalation and necropsied immediately. Spleen, liver, lung and femur were immersion-fixed in 10% neutral buffered formalin, and stained with hematoxylin and eosin using standard protocols. Slides were reviewed by a board-certified pathologist, who confirmed the appearance of metastases and tissue changes.

### 4.11. In Vivo AML Model and Treatment Evaluation with Esperanza Extract

The in vivo model was performed by lateral tail-vein injection (i.v.) of 5 × 10^4^ DA-3/ER-GM cells. To evaluate the effect of treatments on tumor growth, 3 days after tumor cell inoculation, mice were treated with Esperanza extract or PBS (negative control) five times per week. The concentration used was 90 mg/kg body weight of Esperanza extract, which corresponds to 4 doses below the LD_50_, ensuring that the concentration of the extract was not toxic to the animals. Mice were monitored three times per week for body weight, signs of disease and evaluation of hematological parameters. Animals were euthanized by CO_2_ inhalation when welfare losses occurred. At necropsy, spleen, liver, femurs and lungs were obtained and fixed by immersion in 10% of neutral buffered formalin; each tissue was sectioned, fixed and stained with hematoxylin-eosin to evaluate the presence of blasts in these organs using standard protocols. A board-certified pathologist reviewed the slides and confirmed the presence of metastases and tissue changes.

### 4.12. Cytokine Assay

Plasm was prepared from blood obtained by cardiac puncture, and cytokine evaluation was performed using a Cytometric Bead Array (CBA) mouse Th1, Th2, Th17 cytokine kit (BD Biosciences), according to the manufacturer’s instructions. Events were acquired using a FACSAria II flow cytometer (BD Immunocytometry Systems), and the results were subsequently analyzed using FCAP array software version 3.0 (BD Biosciences).

### 4.13. Statistical Analysis

All data are expressed as the mean ± standard error of the mean (SEM). Differences between two groups were determined using the Mann–Whitney U test. Differences among subject groups were evaluated using a one-way ANOVA nonparametric Kruskal–Wallis and Dunn’s post-test for multiple comparisons. GraphPad Prism version 8.1.1 for Mac OS X statistics software (GraphPad Software) was used for the statistical analyses.

## 5. Conclusions

We can consider that the glycolytic alterations induced by Esperanza extract, together with the induction of intracellular ROS levels, and the cytostatic effect, can explain its antitumor activity, and are strongly suggestive that Esperanza extract acts as a drug modulator of tumor metabolism. Activity related to the extract’s metabolite content includes peptides, cinnamic acids, carboxylic acids, fatty acids, and benzoic acids. However, as the role of the extract in the antitumor immune response is not fully elucidated, we are still working on it based on what we have observed in this leukemia model.

## Figures and Tables

**Figure 1 ijms-24-12972-f001:**
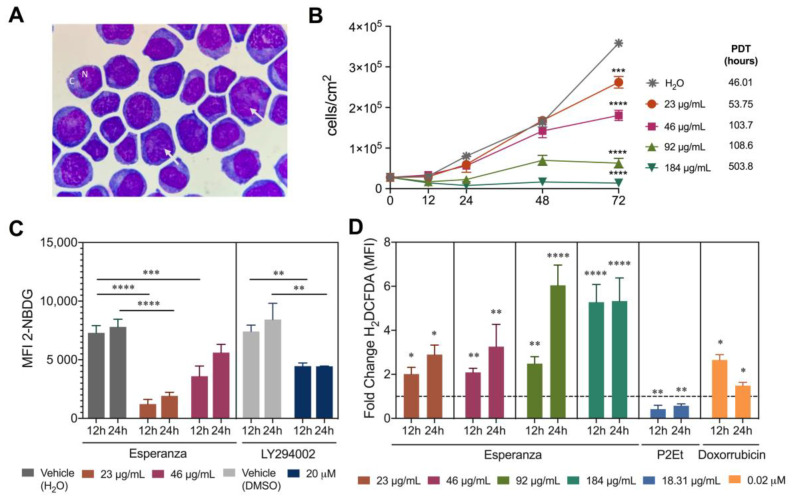
In vitro effect of Esperanza extract in DA-3/ER-GM cells. (**A**) Cytospin preparation of DA-3/ER-GM cells stained using Wright stain prior to photographing using 100× magnification. White arrows indicate poor chromatin; C: cytoplasm; N: nucleus. (**B**) DA-3/ER-GM cell count per cm^2^ after treatment with different concentrations of Esperanza extract or H_2_O (extract vehicle). Population doubling times (PDT) are shown for each treatment. (**C**) Mean fluorescence intensity (MFI) of 2-NBDG in DA-3/ER-GM cells after treatments with Esperanza extract, LY294002 (control of decreased glucose uptake), H_2_O (extract vehicle) or DMSO (LY294002 vehicle) for 12 and 24 h. (**D**) Fold change of H_2_DCFDA MFI after treatments with different concentrations of Esperanza extract, P2Et (anti-oxidant control) or Doxorubicin (pro-oxidant control) for 12 and 24 h. Fold change was determined using the H_2_DCFDA MFI of each treatment relative to its vehicle (the dotted line corresponds to the normalization of the experimental controls). The *p* values were calculated using a one-way ANOVA nonparametric Kruskal–Wallis test with Dunn’s post-test when more than two groups were compared or a Mann–Whitney U test when two groups were compared. * *p* < 0.05, ** *p* < 0.01, *** *p* < 0.001, **** *p* < 0.0001.

**Figure 2 ijms-24-12972-f002:**
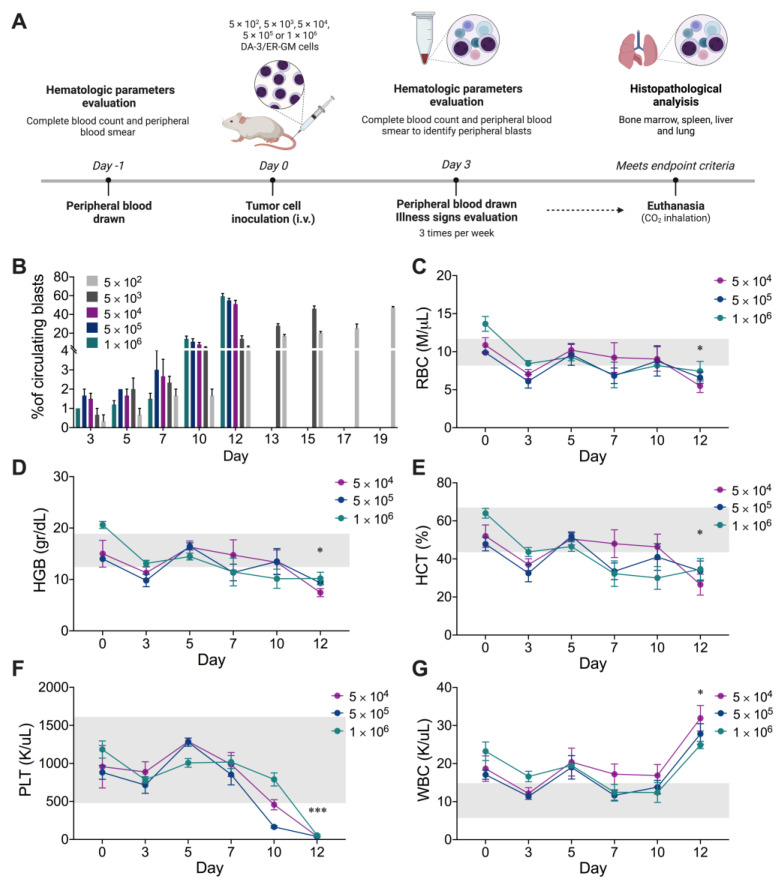
Standardization of the animal model of AML with DA-3/ER-GM cells. (**A**) Experimental design to standardize the AML model with different numbers of DA-3/ER-GM cells. Peripheral blood drawn from the vein of the tail were collected to hematologic parameters evaluation (Day-1). At Day 0, five groups of mice (*n* = 5) were inoculated intravenously (i.v.) by tail vein injection with 5 × 10^2^, 5 × 10^3^, 5 × 10^4^, 5 × 10^5^ or 1 × 10^6^ DA-3/ER-GM cells. All mice were monitored three times per week for body weight, signs of illness and hematological parameters analysis. Mice were euthanized by CO_2_ inhalation and necropsied immediately. Spleen, liver, lung and femur were immersion-fixed in 10% neutral buffered formalin to histopathological analysis. (**B**) Frequency of circulating blasts evaluated with a peripheral blood smear Wright’s stained. (**C**) Red blood cell count (RBC). (**D**) Hemoglobin (HGB). (**E**) Hematocrit (HCT). (**F**) Platelets (PLT). (**G**) White blood cell count (WBC). Hematological parameters (**C**–**G**) were analyzed with a complete blood count using the Mindray BC-5150. In all cases, data are represented as the mean ± SEM. The *p* values were calculated using Mann–Whitney U test and show the differences between day 0 and day 12. * *p* < 0.05, *** *p* < 0.001.

**Figure 3 ijms-24-12972-f003:**
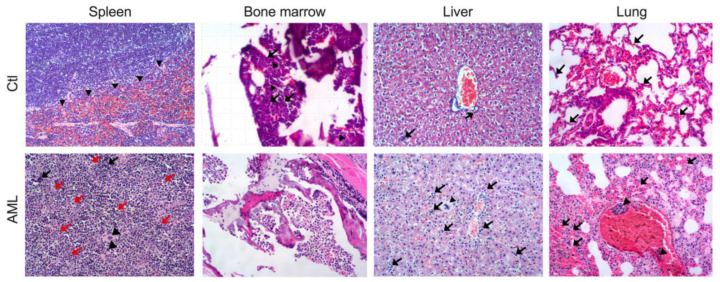
Pathological evidence of leukemia development in animals inoculated with 5 × 10^4^ DA-3/ER-GM cells. Hematoxylin and eosin staining of spleen (40×), bone marrow (40×), liver and lung (40×) derived from healthy control (Ctl, **upper panel**) and leukemic mice (AML, **lower panel**). Foci of mature lymphocytes (black arrows), blasts (red arrows), megakaryocytes (arrowhead).

**Figure 4 ijms-24-12972-f004:**
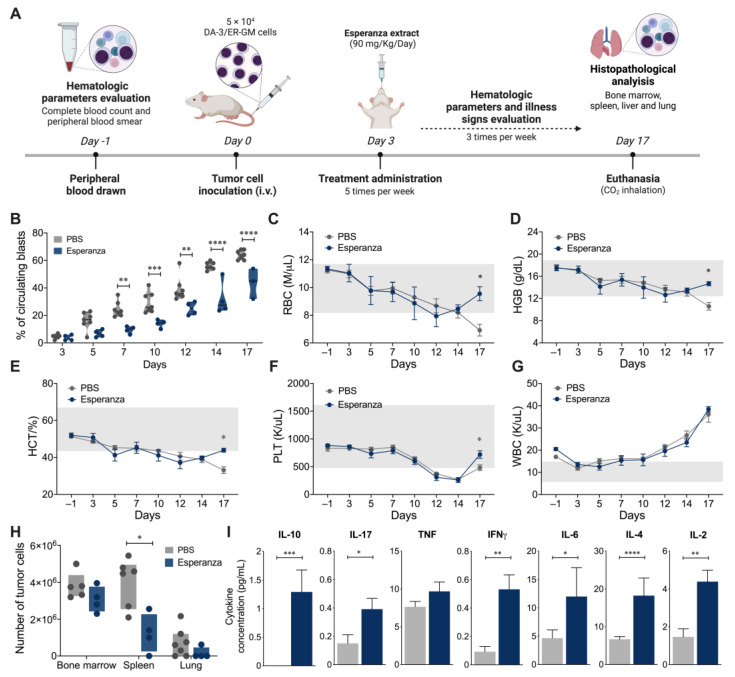
Effect of Esperanza extract in AML mice. (**A**) Experimental design to evaluate the effect of Esperanza extract in AML mice inoculated with 5 × 10^4^ DA-3/ER-GM cells. (**B**) Frequency of circulating blasts evaluated with a peripheral blood smear Wright’s stained. (**C**) Red blood cell count (RBC). (**D**) Hemoglobin (HGB). (**E**) Hematocrit (HCT). (**F**) Platelets (PLT). (**G**) White blood cell count (WBC). Hematological parameters (**B**–**F**) were analyzed with a complete blood count using the Mindray BC-5150. (**H**) Number of tumor cells in bone marrow, spleen and lung. (**I**) Cytokine levels in the plasm of AML mice. In all cases, data are represented as the mean ± SEM. The *p* values were calculated using Mann–Whitney U test. * *p* < 0.05, ** *p* < 0.01, *** *p* < 0.001, **** *p* < 0.0001.

**Figure 5 ijms-24-12972-f005:**
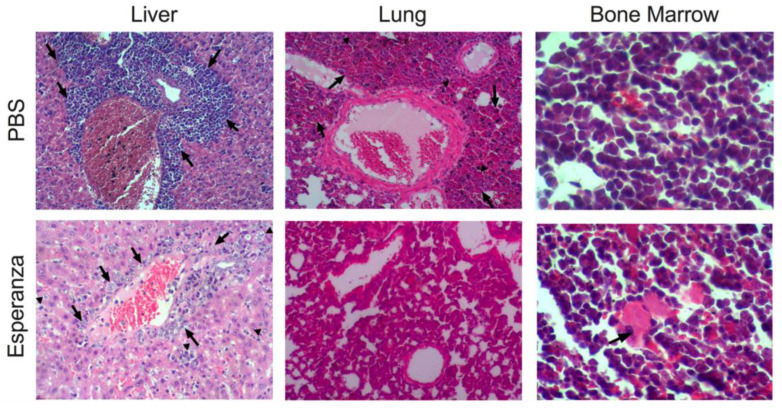
Pathological evidence of the effect of Esperanza extract in AML mice. Hematoxylin and eosin staining of liver (100×), lung (100×), and bone marrow (400×) from PBS (**upper panel**) and Esperanza treated mice (**lower panel**). Liver: immature lymphocytes (arrows); immature lymphocytes in the liver sinusoids (arrowhead). Lung: generalized congestive changes (arrows), areas of alveolar atelectasis (arrowhead). Bone marrow: megakaryocytes (arrows).

**Table 1 ijms-24-12972-t001:** IC50 values of cells after treatment with Esperanza extract.

Cell Lines	IC_50_ (μg/mL)
DA-3/ER-GM	>250
U937	>250
4T1	48 ± 4.8
MCF-7	144 ± 2.8
MDA-MB-468	243.3 ± 9.4
3T3	>250

## Data Availability

The data presented in this study are available in this article (and Appendix A).

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
