# Peer review of "Petiveria alliacea Reduces Tumor Burden and Metastasis and Regulates the Peripheral Immune Response in a Murine Myeloid Leukemia Model"

_ijms, 2023, doi:10.3390/ijms241612972_

Round 1

Reviewer 1 Report

The manuscript “Petiveria alliacea reduces tumor burden and metastasis and regulates the peripheral immune response in a murine myeloid leukemia model” by N. Murillo, P. Lasso, C. Urueña, R. Ballesteros-Ramírez, G. Betancourt, L. Rojas and S. Fiorentino describes the standardization of the model to assess antileukemic effect of a plant extract in vivo and to study the influence of Petiveria alliacea aqueous extract on hematologic and histological indices of BALB mice inoculated with DA-3/ER-GM cells. The effects of P. alliacea aqueous extract on blood indices and histological properties were studied in detail. The conclusions on plausible biochemical mechanisms of action of P. alliacea aqueous extract were drawn. At the same time, the manuscript contains a little, if any information on the composition of extractive cpompounds; it remains unclear, which active chemical compounds provide antiproliferative, immunomodulating action, as well as increase intracellular concentration of reactive oxygen species, and what is the concentration of active compounds in the extract. The comparison between composition and tumor burdening action of aqueous and ethanolic (or ethyl acetate) P. alliacea extracts is absent as well. HPLC chromatogram presented in Supporting information does not shed a light on the composition of extractive compounds. Though the present study was carried out on a good level and maybe of interest to specialists in oncology, phytochemistry and ethnomedicine, it is not consistent with Aims and Scope of International Journal of Molecular Sciences. Given the lack of information on the extract composition and molecular basis of its action (not only discussion of biochemical sequences of the action of P. alliacea extract, but data on specific compounds and their biotargets – proteins, receptors or so on) I cannot recommend the publication of manuscript in IJMS and suggest to consider submitting to the journal of other profile.

Minor flaws:

1.    It is necessary to clarify the structure (or composition) and mode of action of reference compounds: LY294002 (it is named as “LY294004” in Fig. 1 caption), P2Et.

2.    A number of abbreviations should be clarified in the text: 2-NBDG (line 109), ROS (line 114), 2-NBDG MFI (Fig. 1), DTS (line 252), PI (line 422).

3.    Two chemical compounds are named in English erroneously (lines 409, 420).

Author Response

Reviewer 1.

The manuscript “Petiveria alliacea reduces tumor burden and metastasis and regulates the peripheral immune response in a murine myeloid leukemia model” by N. Murillo, P. Lasso, C. Urueña, R. Ballesteros-Ramírez, G. Betancourt, L. Rojas and S. Fiorentino describes the standardization of the model to assess antileukemic effect of a plant extract in vivo and to study the influence of Petiveria alliacea aqueous extract on hematologic and histological indices of BALB mice inoculated with DA-3/ER-GM cells. The effects of P. alliacea aqueous extract on blood indices and histological properties were studied in detail. The conclusions on plausible biochemical mechanisms of action of P. alliacea aqueous extract were drawn. At the same time, the manuscript contains a little, if any information on the composition of extractive compounds; it remains unclear, which active chemical compounds provide antiproliferative, immunomodulating action, as well as increase intracellular concentration of reactive oxygen species, and what is the concentration of active compounds in the extract. The comparison between composition and tumor burdening action of aqueous and ethanolic (or ethyl acetate) P. alliacea extracts is absent as well. HPLC chromatogram presented in Supporting information does not shed a light on the composition of extractive compounds. Though the present study was carried out on a good level and maybe of interest to specialists in oncology, phytochemistry and ethnomedicine, it is not consistent with Aims and Scope of International Journal of Molecular Sciences. Given the lack of information on the extract composition and molecular basis of its action (not only discussion of biochemical sequences of the action of P. alliacea extract, but data on specific compounds and their biotargets – proteins, receptors or so on) I cannot recommend the publication of manuscript in IJMS and suggest to consider submitting to the journal of other profile.

Response:

We appreciate your comment and agree with your assessment. Fortunately, parallel work on the chemical characterization of the Esperanza extract was in progress and, therefore, we include these results in the article and complement our discussion. We consider that with these new results we complement our article and it could be considered within the scope of the journal.

Minor flaws:

1. It is necessary to clarify the structure (or composition) and mode of action of reference compounds: LY294002 (it is named as “LY294004” in Fig. 1 caption), P2Et.

Response: Following the evaluator's recommendation, we include additional information on the LY294002 inhibitor and the P2Et extract. In addition, we make the correction mentioned in Figure 1.

2. A number of abbreviations should be clarified in the text: 2-NBDG (line 109), ROS (line 114), 2-NBDG MFI (Fig. 1), DTS (line 252), PI (line 422).

Response: Following the reviewer's assessment, the recommended changes were made.

3. Two chemical compounds are named in English erroneously (lines 409, 420).

Response: The names of the chemical compounds were corrected.

Reviewer 2 Report

Despite significant progress in the field of creating highly effective drugs for the treatment of oncological diseases, the need for medicine in the search for new more effective low-toxic drugs remains very relevant due to the phenomenon of multiple drug resistance. The team of authors performed an interesting study on the alcoholic extract of Petiveria alliacea, which, as shown, has a significant antitumor effect in vitro on breast adenocarcinoma and other leukemic cells, as well as the ability to regulate energy metabolism.

In addition, the in vitro and in vivo effects of Petiveria alliacea (Esperanza) extract in a mouse model of AML with DA-3/ER-GM cells have been studied. As a result, the extract was shown to modulate tumor metabolism by reducing glucose consumption and increasing the amount of reactive oxygen species, which leads to a decrease in cell proliferation. Animals treated with Esperanza extract had a lower percentage of circulating blast cells in the peripheral blood, higher values of hemoglobin, hematocrit and platelets, less tumor cell infiltration in the spleen, and greater production of cytokines compared to the control group.

The study is very well designed and due to the combination of in vitro and in vivo methodologies is very interesting. Meanwhile, there are a number of comments and concerns:

1. The composition of the Petiveria alliacea extract has been studied at different times, so the authors should list, and ideally study from the individual influence of the major and minor components that could lead to the observed effects. For scientific research, the use of extracts is permissible, but if we talk about its introduction as a drug into medical practice, then due to the lack of the possibility of standardization and control of the constancy of the composition of the extract, its use is minimized. These aspects should be discussed separately in the article. To attract more researchers, it may be necessary to insert the chemical formulas of the major components of the Petiveria alliacea extract into the article.

2. A number of methods in the study of the in vitro effect of the extract on tumor cells were carried out using flow cytometry methods, but I did not find cytometric rafts and their description either in the article or in the appendices. I think they should be brought.

3. What substances were used as negative and positive controls? How was the statistical processing of experiments in vitro and in vivo carried out? What is the selectivity index of the effect on tumor cells compared to normal for the studied extract of Petiveria alliacea?

Author Response

Reviewer 2.

Despite significant progress in the field of creating highly effective drugs for the treatment of oncological diseases, the need for medicine in the search for new more effective low-toxic drugs remains very relevant due to the phenomenon of multiple drug resistance. The team of authors performed an interesting study on the alcoholic extract of Petiveria alliacea, which, as shown, has a significant antitumor effect in vitro on breast adenocarcinoma and other leukemic cells, as well as the ability to regulate energy metabolism.

In addition, the in vitro and in vivo effects of Petiveria alliacea (Esperanza) extract in a mouse model of AML with DA-3/ER-GM cells have been studied. As a result, the extract was shown to modulate tumor metabolism by reducing glucose consumption and increasing the amount of reactive oxygen species, which leads to a decrease in cell proliferation. Animals treated with Esperanza extract had a lower percentage of circulating blast cells in the peripheral blood, higher values of hemoglobin, hematocrit and platelets, less tumor cell infiltration in the spleen, and greater production of cytokines compared to the control group.

The study is very well designed and due to the combination of in vitro and in vivo methodologies is very interesting. Meanwhile, there are a number of comments and concerns:

  1. The composition of the Petiveria alliacea extract has been studied at different times, so the authors should list, and ideally study from the individual influence of the major and minor components that could lead to the observed effects. For scientific research, the use of extracts is permissible, but if we talk about its introduction as a drug into medical practice, then due to the lack of the possibility of standardization and control of the constancy of the composition of the extract, its use is minimized. These aspects should be discussed separately in the article. To attract more researchers, it may be necessary to insert the chemical formulas of the major components of the Petiveria alliacea extract into the article.

Response: In accordance with the suggestion, we included the characterization results that we were working on in parallel in the laboratory and expanded the discussion.

  1. A number of methods in the study of the in vitro effect of the extract on tumor cells were carried out using flow cytometry methods, but I did not find cytometric rafts and their description either in the article or in the appendices. I think they should be brought.

Response: Following the suggestion, we included Supplementary Figure 2 with representative flow cytometric histograms from in vitro assays.

  1. What substances were used as negative and positive controls? How was the statistical processing of experiments in vitro and in vivo carried out? What is the selectivity index of the effect on tumor cells compared to normal for the studied extract of Petiveria alliacea?

Response: We appreciate each of the requests and then we respond to each of them:

  • In the legends, the information of the positive and negative controls was extended.
  • The statistical information was reviewed and adjusted in the legends of the figures.
  • The IC50 of the Esperanza extract in normal 3T3 fibroblasts was greater than 250 μg/ml and was included it in Table 1 as a reference, however, for this value, we did not calculate the selectivity index.

Round 2

Reviewer 1 Report

Authors improved the manuscript by means of the study of chemical composition of the extract by LC and GC in a combination with HRMS. Scientific soundness increased significantly after additions in the text and in Supporting information.

The only question raised is related to Supplementary Table 1. In the case of analysis of compounds by GC-MS pre-column derivatization was carried out (see the Manuscript, section 4.4.2). So, in the Table should be listed the values of moleciular weight and retention time of silylated compounds. It should be clarified; I suggest to divide Supplementary Table 1 in two separate Tables (LC-MS and GC-MS) and specify that compounds were identified after appropriate derivatization (as a footnote or in the table itself).

I highlighted several little remarks in the text (please, find the file attached). After fixing the mentioned issues article can be published in IJMS.

Author Response

Dear reviewer,
We appreciate your quick response and as suggested, all the mentioned changes were made including the division of the supplementary table.

Reviewer 2 Report

I am completely satisfied with the answers and the edits made of the authors of the article. I think that the article can be accepted for publication.

Author Response

Dear reviewer,
We appreciate your prompt and affirmative response.